# Is Metformin Use Associated with a More Favorable COVID-19 Course in People with Diabetes?

**DOI:** 10.3390/jcm13071874

**Published:** 2024-03-24

**Authors:** Giovanni Antonio Silverii, Carlo Fumagalli, Renzo Rozzini, Marta Milani, Edoardo Mannucci, Niccolò Marchionni

**Affiliations:** 1Experimental and Clinical Biomedical Sciences “Mario Serio” Department, University of Florence, 50134 Florence, Italygiovanniantonio.silverii@unifi.it (G.A.S.); 2Department of Advanced Medical and Surgical Sciences, Università degli Studi della Campania “Luigi Vanvitelli”, 80138 Naples, Italy; 00.carlo@gmail.com; 3Department of Internal Medicine and Geriatrics, Fondazione Poliambulanza Istituto Ospedaliero, 25124 Brescia, Italy; renzo.rozzini@iol.it; 4Experimental and Clinical Medicine Department, University of Florence, 50134 Florence, Italy; niccolo.marchionni@unifi.it

**Keywords:** COVID-19 mortality, COVID-19 MRS, Diabetes Mellitus, metformin

## Abstract

**Background:** Diabetes Mellitus (DM) has been associated with a higher Coronavirus disease-19 (COVID-19) mortality, both in hospitalized patients and in the general population. A possible beneficial effect of metformin on the prognosis of COVID-19 has been reported in some observational studies, whereas other studies disagree. **Methods:** To investigate the possible effect of metformin on COVID-19 in-hospital mortality, we performed a retrospective study that included all SARS-CoV-2-positive patients with DM who were admitted to two Italian hospitals. In order to adjust for possible confounders accounting for the observed reduction of mortality in metformin users, we adopted the COVID-19 Mortality Risk Score (COVID-19 MRS) as a covariate. **Results:** Out of the 524 included patients, 33.4% died. A binomial logistic regression showed that metformin use was associated with a significant reduction in case fatality (OR 0.67 [0.45–0.98], *p* = 0.039), with no significant effect on the need for ventilation (OR 0.75 [0.5–1.11], *p* = 0.146). After adjusting for COVID-19 MRS, metformin did not retain a significant association with in-hospital mortality [OR 0.795 (0.495–1.277), *p* = 0.342]. **Conclusions:** A beneficial effect of metformin on COVID-19 was not proven after adjusting for confounding factors. The use of validated tools to stratify the risk for COVID-19 severe disease and death, such as COVID-19 MRS, may be useful to better explore the potential association of medications and comorbidities with COVID-19 prognosis.

## 1. Introduction

Four years after the first onset of COVID-19, the disease still seems to exert a detrimental effect on life expectancy in many countries [1], despite an enormous improvement in its prognosis following mass vaccination [2] and the development of antiviral drugs [3]. The risk for a severe form of COVID-19, even in those who received vaccination, is significantly reduced, but it may still not be negligible in patients with comorbidities [4]. Among the more frequent chronic diseases, Diabetes Mellitus (DM) has been associated with a 14–32% higher possibility of a severe course of disease [5] and a higher mortality for Coronavirus disease-19 (COVID-19), both in hospitalized patients [6] and in the general population [7], especially in those with a longer disease duration [8]. This finding is in line with previous observations which established DM as a risk factor for a more severe course of several infectious illnesses [9], including a higher risk for influenza-related hospitalization and death [10] and a higher risk for mortality for other diseases caused by coronaviruses, such as Severe Acute Respiratory Syndrome (SARS-CoV-1) [11], or Middle East Respiratory Syndrome Coronavirus (MERS-CoV) [12]. Furthermore, the hesitancy toward the COVID-19 vaccine is not negligible in people with DM, especially in those with further risk factors such as worse glycemic control and obesity [13], potentially increasing the harmful potential of COVID-19 in people with Diabetes Mellitus [14]. Possible mechanisms explaining the association between DM and an increased risk for COVID-19 severity may be found in a higher Angiotensin-converting enzyme 2 (ACE2) expression in patients with DM [15], higher levels of TNFα, and lower levels of the anti-inflammatory cytokine, IL-10 [16].

Glucose-lowering drugs have also been investigated for their possible influence in modulating the prognosis of COVID-19 in people with Diabetes Mellitus [17]. In particular, a possible beneficial effect of metformin has been reported with respect to mortality or hospitalization for COVID-19 in observational studies [18], both in population-based epidemiological studies [7,19] and hospitalized patients [20]. However, some studies provided conflicting results, as they did not detect any difference in COVID-19 prognosis in patients on metformin [19,21]. Several mechanisms have been proposed for the potential benefit of metformin during COVID-19 infection. The well-known glucose-lowering effect of metformin, by improving glycemia, may also help to reduce the risk of a severe disease course, as proposed by Ceriello et al. [22]; accordingly, a beneficial effect may be harbored by the reduction in body weight [23] and the improvement in insulin resistance [24]. In addition, metformin has been under investigation as an antiviral agent since its first discovery, having shown some efficacy when used for the treatment of influenza [25], and it has also shown in vitro activity in reducing SARS-CoV-2 titers, although with heterogeneous results [26]. Furthermore, the drug has also shown protection against lipopolysaccharide-induced lung injury in mice inoculated with SARS-CoV-2 [27], and it may also have a direct effect on the endosomal Na^+^/H^+^ exchanger, which in turn may lead to an increased cellular pH, and thus interfere with the endocytic cycle of the virus [28]. Metformin enhances the adenosine monophosphate-activated kinase (AMPK), which in turn phosphorylates ACE2, thereby inhibiting the penetration of the virus. Other possible mechanisms have been involved in the possible effect of metformin on COVID-19, as metformin has been observed to modulate the acute and chronic inflammatory response; metformin activates AMPK in primary macrophages and in macrophage-like cell lines, a mechanism which is independent of HIF1-α and IL-10. On the other hand, metformin especially lowers interleukin-1β and interleukin-6 levels [29], showing thus anti-inflammatory properties [30], whereas the inhibition of the mTOR pathway may play a role in the prevention of immune hyperactivation [31]. Furthermore, it may ameliorate the neutrophil’s response to infection, preventing excessive activation [32]. Metformin has also been associated with the inhibition of the electron transport mitochondrial complex 1, leading to the suppression of mitochondrial reactive oxygen species (ROS) signaling, the prevention of ROS formation, and ROS-mediated interleukin-6 release [33]. Furthermore, metformin reduces platelet activation and the risk for thrombosis [34], which is a possible contributor to severe COVID-19 disease. Also, the inhibition of dipeptidyl peptidase 4 (DPP4) may also be theoretically implied; the DPP4 Sitagliptin has been proposed as possibly associated with a more favorable prognosis of COVID-19 [35], following the observation that the inhibition of DPP4 leads to reduced production of inflammatory cytokines in pulmonary alveoli cells in the lung, potentially preventing the “cytokine storm” induced by the spike surface antigen of SARS-CoV-2 during the severe forms of the disease [36]. Metformin, although to a lesser extent than that of DPP4 inhibitors, has shown an inhibiting effect on DPP4 [37].

A meta-analysis of 20 observational studies investigating the mortality risk of COVID-19 for diabetic patients taking metformin showed that previous home use but not in-hospital use of metformin was associated with a statistically significant reduced mortality risk (pooled OR = 0.62, 95% CI, 0.50–0.76), in Europe and North America [38], but the result was affected by a high heterogeneity (I^2^ = 77.6%), likely because in this analysis, the odds ratio derived from studies adjusting for different confounding factors, and those derived from studies in which unadjusted odds ratios were performed, were all included and meta-analyzed together. On the other hand, in a randomized trial, the early use of metformin in ambulatory patients with COVID-19 did not show any clinical advantage over placebo in the incidence of hospitalization, viral discharge, or other secondary clinical outcomes [39], whereas another randomized clinical trial, although not showing any superiority over placebo in reducing the risk for a severe course of acute SARS-CoV2 infection, such as hypoxemia, emergency department referral, hospitalization, or death [40], showed a reduction in the incidence of long-COVID in overweight or obese patients [41]. The possible effect of metformin on COVID-19 course remains controversial; the aim of our study was, therefore, to investigate the possible effect of metformin on COVID-19 case fatality in hospitalized patients.

## 2. Materials and Methods

### 2.1. Study Population

We performed a retrospective observational cohort study on a consecutive series of hospitalized patients. We included all the SARS-CoV-2 positive patients aged > 18 years who were diagnosed with DM at baseline and who were admitted to two Italian tertiary hospitals located in Northern and Central Italy (Poliambulanza Hospital, Brescia and Careggi University Hospital, Florence) from 22 February 2020 (date of first admission in Brescia) to 30 April 2021. Patients were included if the concomitant medication and comorbidities and their vital status 60 days after discharge from the hospital were available in their medical charts.

### 2.2. Endpoints

The principal endpoint was in-hospital death, which was defined as death occurring during hospitalization or less than 60 days after discharge. The need for assisted ventilation was a secondary endpoint, as well as the need for oxygen, and the hospitalization duration.

### 2.3. Data Collection

All data on patients’ medical history, physical parameters, biochemical exams, and concomitant medications, as well as on the prespecified outcomes, were retrieved from electronic patient records. Furthermore, upon hospital discharge, all patients were enrolled in a prospective follow-up study, whereby healthcare assistants would make pre-scheduled phone calls at 1, 3, 6, 9, and 12 months to enquire about survival and symptoms burden. Patients were later invited to participate in an observational trial aimed at determining the persistence of symptoms post-discharge. All records were entered in a REDCap (Research Electronic Data Capture, REDCap 8.11.6-© 2024 Vanderbilt University), which was used as a source to retrieve data on mortality. All healthcare assistants had received prior training in order to acquire and record data reliably [42].

In keeping with statements by the Italian Regulatory Authorities for observational studies, the Ethical Committees of both hospitals approved data collection and granted a waiver of informed consent from study participants.

### 2.4. Statistical Analysis

Data were synthesized as median and quartiles. Nonparametric analyses (Pearson Chi-Square) were performed to investigate any significant differences between medians for continuous parameters, whereas, for dichotomic variables, a binomial logistic regression was made. If a variable was significantly associated with an increased occurrence of an outcome, a multivariate analysis was designed, including all the parameters associated with the outcome in the bivariate analysis. In order to adjust for possible confounders accounting for the observed reduction of mortality in metformin users, we adopted the COVID-19 Mortality Risk Score (COVID-19 MRS) as a covariate [43] in our multivariate analysis model. The COVID-19 MRS, which is based on age, number of chronic diseases, respiratory rate, the ratio of arterial oxygen partial pressure (PaO_2_) to fractional inspired oxygen (FiO_2_), serum creatinine, and platelet count, is a highly accurate tool in stratifying patients at low, intermediate, and high risk of in-hospital death from COVID-19, as emerged in a study aimed at this validation [43]. All the analyses were performed with IBM SPSS Statistics Software 29.0.2.0. (20).

## 3. Results

We included 524 patients in the analysis. Of those, 220 (44%) were in therapy with metformin. At baseline, 70.4% of the sample were males, 31.6% were obese, 63.5% suffered from hypertension, 13.8% had a stage 4 or 5 CKD, and 27.3% were cigarette smokers; furthermore, the enrolled patients had a median age of 74 years (64; 78 interquartile range), a median number of four comorbidities, and a median number of seven medications. The characteristics of the patients included are reported in Table 1. Patients who were in therapy with metformin were significantly younger than those who were not on metformin (72 versus 75 years of median age, *p* = 0.005); furthermore, metformin users had a non-significantly lower prevalence of stroke and cardiovascular disease, and a non-significantly higher prevalence of obesity and hypertension, with significantly higher use of ACE-I or ARBS (39.6 versus 26.3% of users, *p* = 0.014), whereas the median number of medications and comorbidities was not significantly different. The median duration of hospitalization was seven days, without a significant difference between those in therapy with or without metformin. The vast majority of patients (89.9%) needed oxygen, without any difference between those on therapy and those not on therapy with metformin. Those who needed assisted ventilation were 30% in the whole sample; there was no significant observed effect of metformin on the need for ventilation (OR 0.75 [0.5–1.11, 95% CI], *p* = 0.146) in a bivariate analysis. Overall, 74 patients (33.4%) died during hospitalization or within 60 days after discharge. The COVID-19 MRS was confirmed as a strong predictor of mortality (r = 0.42, *p* < 0.0001, with a 3.5 OR [2.4–5.6, 95% CI] for each increase in the MRS score category). A significantly greater proportion of patients on metformin therapy was considered at a low or intermediate MRS risk (34.8% and 36% of metformin users were at low or intermediate risk, respectively, versus 26.6% and 33.2% of those not on metformin), whereas a greater proportion of patients not on metformin therapy were classified as at high MRS risk (40.3% of those not on metformin, versus 29.3% of those on metformin). Age (OR 1.97 [1.62–2.37, 95% CI], *p* < 0.0001) and stage 4 or 5 CKD (OR 1.97 [1.62–2.37, 95% CI], *p* < 0.0001) were the strongest single predictors, among the components of the MRS score; furthermore, the presence of previous heart disease, which was not included in the score, was strongly associated with a higher mortality in people with diabetes (OR 2.2 [1.40–3.20, 95% CI], *p* < 0.0001). On the other hand, obesity (OR 0.79 [0.50–1.25, 95% CI], *p* = 0.31), hypertension (OR 1.19 [0.98–1.50, 95% CI], *p* = 0.08), male sex (OR 0.73 [0.48–1.1, 95% CI], *p* = 0.729), cigarette smoking (OR 1.5 [0.90–2.7, 95% CI], *p* = 0.114), number of comorbidities (OR 1.1 [0.97–1.23, 95% CI], *p* = 0.144), number of medications (OR 1.02 [0.91–1.14], 95% CI, *p* = 0.716), being on therapy with ACE inhibitors or Angiotensin Receptor blockers (ACE-I/ARBS, OR 0.78 [0.53–1.14], 95% CI, *p* = 0.196) did not show a significant correlation with mortality (Figure 1). Of the 220 patients who were on metformin therapy, 60 (27.3%) died, whereas the proportion of deceased patients who were not on metformin was 36% (99 out of 275 patients). A binomial logistic regression showed that metformin use was associated with a significant reduction in case fatality (OR 0.67 [0.45–0.98], 95% CI, *p* = 0.039, Figure 1). After adjusting for COVID-19 MRS, metformin use did not retain a significant association with in-hospital mortality (OR 0.795 [0.495–1.277] (OR 1.1 [0.97–1.23, 95% CI], *p* = 0.144), *p* = 0.342, Figure 2), whereas the MRS score (OR 2.96 [2.12–4.13], 95% CI, *p* < 0.0001) and the presence of a previous heart disease (OR 1.70 [1.03–2.79], 95% CI, *p* = 0.037) both retained significance with in-hospital mortality in a multivariate model (Figure 2).

In our sample, 33.3% of patients were prescribed Ritonavir + Lopinavir or Darunavir + Cobicistat antiviral medication. Patients on metformin had a significantly higher proportion of antiviral prescriptions (Table 1). A bivariate analysis showed a non-significant correlation between antiviral medications and in-hospital mortality (OR 2.44 [0.97–6.14], 95% CI, *p* = 0.06). Furthermore, when adding antiviral use to the multivariate model, together with metformin, previous heart disease, and COVID-MRS, neither antiviral use (OR 1.89 [0.64–5.62], 95% CI, *p* = 0.25) nor metformin use (OR 0.71 [0.27–1.9], 95% CI, *p* = 0.50), showed any significant correlation with in-hospital mortality.

## 4. Discussion

In our study, a beneficial effect of metformin on mortality for COVID-19 was apparent in the bivariate analysis, but it was not confirmed after adjusting for possible confounding factors. Our findings are in line with some of the previous observational studies [19,21] and with a previous observational study also reporting that an observed significantly lower proportion of tracheal intubation and death from COVID-19 in metformin users observed in a bivariate analysis became no longer significant when performing multivariate analyses [44]. On the other hand, our results are in contrast with recent papers comparing different treatments for diabetes with respect to COVID-19 courses [45]. The discrepancy between different studies in the observed effect of metformin may be explained by differences in the included population characteristics, such as patients’ functional status or severity of disease, which in turn may be influenced by the course of the epidemic and by the clinical care beds availability. Furthermore, the difference in the design of the analysis and the different moderators adopted to build the adjusted models in each study may at least in part explain the observation of conflicting results; on the other hand, a distortion determined by unidentified confounders can never be excluded in observational studies. The issue of metformin effect on COVID-19 course may have relevant clinical implications in the management of diabetes during COVID-19 disease [46] because metformin is usually suspended in patients hospitalized for pneumonia or respiratory insufficiency due to the potentially increased risk of lactic acidosis [47]; it is, therefore, essential to provide evidence of a specific effect of metformin on COVID-19. Many molecular mechanisms have been advocated to explain a possible effect of metformin on COVID-19, such as its well-established insulin-sensitizing action or possible anti-inflammatory or antiviral properties [48], in a similar fashion to the possible effect that has been proposed for metformin on influenza-related complications [49]. Observational studies and randomized clinical trials, however, have so far provided controversial results [30,39]. Our study, on the other hand, suggests that the use of metformin, rather than playing a direct role in the COVID-19 course, may be more frequent in clinical conditions associated with a better prognosis of COVID-19, thus producing an apparent beneficial effect. Metformin, in fact, is not indicated in people with severe to end-stage kidney disease, corresponding to a glomerular filtration rate lower than 30 mL/min [50], which is, in turn, a condition that, in our study, has been associated with higher mortality for COVID-19. Furthermore, in patients with severe heart failure and respiratory insufficiency, which are conditions at a higher risk for severe COVID-19, metformin is also not indicated [51]. The exclusion of such cases in the metformin arm may lead to a selection bias, and this may contribute to explaining the observed lower case fatality ratio for COVID-19 in patients in therapy with metformin. Notably, the association of obesity with mortality among people with diabetes, which has been reported in the literature [21], has not been confirmed in our study. On the other hand, Ritonavir + Lopinavir and Darunavir + Cobicistat, which were used as antiviral medications, did not show an association with overall survival, which is in line with recent literature [52,53].

In our study, we adopted the COVID-19 risk score as a possible confounder, which includes the severity of COVID-19 symptoms, such as respiratory rate, FIO_2_, and PaO_2_, and it was already validated in our sample [43]. The metformin user group had a significantly lower proportion of patients at a higher risk for severe COVID-19, according to the COVID-19 MRS; therefore, an adjusted analysis including the COVID-19 MRS score seems to overcome possible confounding factors, ruling out a direct effect of metformin in COVID-19. This is a strength of our study, as the use of prognostic scores, such as COVID-19 MRS, may contribute to explaining the apparent effect of metformin on COVID-19 disease. Furthermore, the COVID-19 MRS is not operator-dependent, and it can easily be adapted to objectively predict the possibility of an unfavorable outcome in patients with COVID-19.

A major limitation of our study, on the other hand, is the absence of information on blood glucose or glycated hemoglobin levels, which may be confounding factors, given that a worse glycemic control has been associated with a more severe COVID-19 course [8,54,55]. Furthermore, no data on the prevalence of microvascular complications, a condition which has also been linked to a worse prognosis of COVID-19 [55], were available in our retrospective study, as well as no data on the ethnicity and socioeconomic status of the patients. Moreover, data on the availability of healthcare were not available for our sample, a factor that has been proven to influence COVID-19 prognosis [56]. A further major limitation is that we could not retrieve information on the cause of death for the included patients.

In conclusion, in our observational study, hospitalized patients with COVID-19 who were on therapy with metformin showed a significantly lower case fatality, but the association between metformin use and a more favorable prognosis of COVID-19 was not confirmed in a multivariate analysis. The use of validated tools to stratify the risk for COVID-19 severe disease and death, such as the COVID-19 MRS, may be useful to better explore the potential association of medications and comorbidities with COVID-19 prognosis in observational studies.

## Figures and Tables

**Figure 1 jcm-13-01874-f001:**
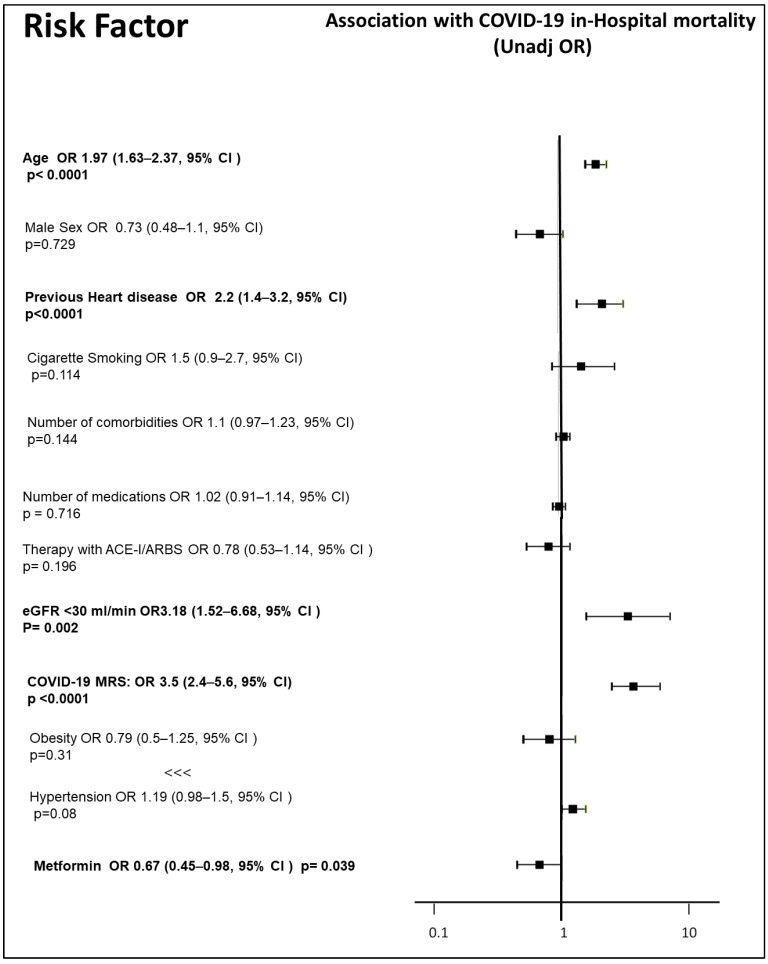
Risk factors associated with in-hospital mortality for COVID-19: univariate analysis. Significant association results are shown in bold.

**Figure 2 jcm-13-01874-f002:**
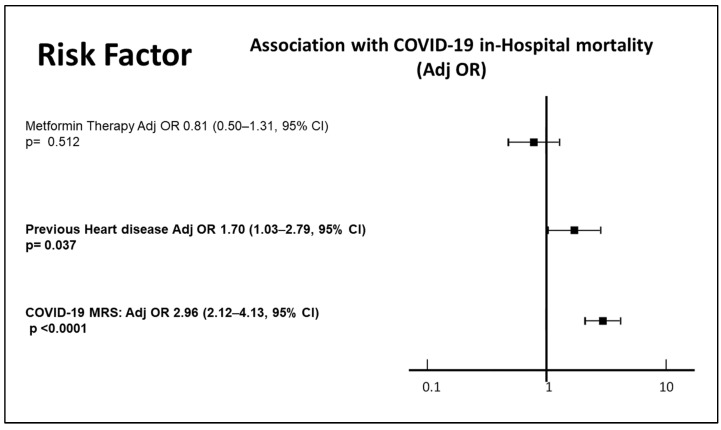
Risk factors associated with in-hospital mortality for COVID-19: multivariate analysis. Significant association results are in shown in bold.

**Table 1 jcm-13-01874-t001:** Patients’ characteristics: differential characteristics between patients who were on therapy with metformin and those who were not. Here, not otherwise specified, data are expressed as median (quartiles) with 95% interquartile range. Characteristics were considered significantly different between the two groups if *p* was lower than 0.05. BMI = Body Mass Index; ACE-I = Angiotensin Converting Enzime-Inhibitors; ARBS = Angioten Receptor Blockers; eGFR = estimated Glomerular Filtration Rate; Hb = Hemoglobin.

Variable	All (*n* = 524)	Metformin Users (*n* = 220)	Not Metformin Users (*n* = 304)	*p*
Age (years)	74 (66; 81)	72 (64; 78)	75 (68; 82)	0.005
Male gender (%)	70.4	69.5	71.3	0.69
Cigarette smokers (%)	27.3	28.2	23.7	0.47
Previous heart disease (%)	46.5	42.7	52.1	0.062
Previous stroke (%)	9.8	7.5	13.2	0.30
Respiratory Rate (breath/min)	20 (18; 28)	20 (18; 28)	20 (18; 29.5)	0.82
Number of comorbidities	4 (3; 5)	4 (3; 5)	4 (3; 5)	0.11
Number of medications	7 (4; 10)	7 (3; 9.5)	8 (5; 10)	0.47
Obesity (BMI > 30 kg/m^2^) (%)	31.6	32.2	30.9	0.91
Hypertension (%)	63.5	68.8	61.4	0.12
ACE-I/ARBS therapy (%)	36.8	39.6	26.3	0.014
eGFR < 30 mL/min (%)	13.8	10.9	16.1	0.27
Hospitalization duration (days)	7 (4; 13)	8 (4; 12)	7 (4; 13)	0.29
Hb on admission (g/dL)	12.5 (11.2–13.7)	13.8 (11.4–13.8)	12.5 (11–13.9)	0.231
Creatinine on admission (mg/dL)	1.12 (0.73–1.61)	1.03 (0.81–1.39)	1.23 (0.86–1.85)	0.022
Need for Oxygen therapy (%)	89.9	88.6	93	0.28
Need for assisted ventilation (%)	30	26.2	32.2	0.16
Death (%)	33.4	27.3	36	0.042
Antiviral use (%)	33.3	41.5	17.7	0.02
COVID-19 Mortality Risk score:				0.033
Low, <10 points (%)	30	34.8	26.6	
Intermediate, 11–13 points (%)	33.7	36	33.2	
High, ≥14 points (%)	36.4	29.3	40.3	

## Data Availability

The data presented in this study are available on request from the corresponding author, due to privacy reasons.

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
