# Peer review of "Is Metformin Use Associated with a More Favorable COVID-19 Course in People with Diabetes?"

_jcm, 2024, doi:10.3390/jcm13071874_

Round 1

Reviewer 1 Report

Comments and Suggestions for Authors

The study titled "Is Metformin use associated with a more favorable COVID-19 course in people with Diabetes?" authored by Silverii et al., seeks to underscore the potential beneficial effects of metformin in the context of COVID-19. Overall, the study provides some insights into the association between metformin therapy and mortality in COVID-19 patients, however, there are several limitations and potential loopholes that need to be considered when interpreting the findings.

1.      The study includes 524 patients, which is a small sample size for such study. Furthermore, the study does not specify how patients were selected, which could introduce bias. Additionally, the study lacks information about the demographic characteristics of patients beyond what's mentioned (e.g., socioeconomic status, ethnicity), which could impact the generalizability of the findings.

2.      The study mentions several baseline characteristics of the patients, including age, sex, comorbidities, and medication use. However, it's unclear whether the researchers controlled for all potential confounding variables adequately. For instance, factors like socioeconomic status, access to healthcare, and severity of COVID-19 symptoms could influence both metformin use and mortality outcomes.

3.      The primary outcome of interest is mortality. However, the study doesn't provide information on the causes of mortality or whether there were differences in causes of death between the metformin and non-metformin groups. This information would be crucial for understanding the mechanism behind any observed effects.

4.      The study mentions that mortality was assessed during hospitalization or within 60 days after discharge. However, it's unclear how patients were followed up after discharge and whether all deaths within this period were captured reliably.

Author Response

    1. The study includes 524 patients, which is a small sample size for such study. Furthermore, the study does not specify how patients were selected, which could introduce bias.

    We thank the reviewer for her/his criticism. As stated in the Materials and Methods section, first paragraph: “We included all the Sars COV-2 positive patients aged >18 years which were diagnosed with DM at baseline, and which were admitted to two Italian tertiary hospitals located in Northern and Central Italy (Poliambulanza Hospital, Brescia; and Careggi University Hospital, Florence)”, therefore, no selection was made. We added a sentence in this section, to further clarify this point.

    Additionally, the study lacks information about the demographic characteristics of patients beyond what's mentioned (e.g., socioeconomic status, ethnicity), which could impact the generalizability of the findings.

    1. The study mentions several baseline characteristics of the patients, including age, sex, comorbidities, and medication use. However, it's unclear whether the researchers controlled for all potential confounding variables adequately. For instance, factors like socioeconomic status, access to healthcare, and severity of COVID-19 symptoms could influence both metformin use and mortality outcomes.

    We thank the reviewer for her/his remarks. Data on socioeconomic status, ethnicity and access to healthcare were not available. We have now added this limitation to the discussion section, third paragraph. As for the severity of  COVID-19, symptoms, it was included in the COVID-19 MRS, which, as stated Method section, third par. included respiratory rate, ratio of arterial oxygen partial pressure (PaO2) to fractional inspired oxygen (FiO2), serum creatinine and platelet count, and it had already been validated in the observed sample. We added a further sentence, in the discussion section, second paragraph,  for greater clarity.

    1. The primary outcome of interest is mortality. However, the study doesn't provide information on the causes of mortality or whether there were differences in causes of death between the metformin and non-metformin groups. This information would be crucial for understanding the mechanism behind any observed effects.

    We agree with the reviewer that this is a major limitation of our study. Unfortunately, this information could not be retrieved. We added this limitation to the discussion section, third paragraph.

    The study mentions that mortality was assessed during hospitalization or within 60 days after discharge. However, it's unclear how patients were followed up after discharge and whether all deaths within this period were captured reliably.

    We thank the reviewer for her/his careful criticism. Upon hospital discharge, all patients were enrolled in a prospective follow up study whereby healthcare assistants would make pre-scheduled phone calls at 1,3,6,9 and 12 months from hospital discharge.  to enquire about survival and symptoms burden. Patients were later invited to participate to a observational trial aimed at determining persistence of symptoms post-discharge. All records were entered in a REDCap (Research Electronic Data Capture, REDCap 8.11.6 - © 2021 Vanderbilt University). All healthcare assistants had received prior training in order to acquire and record data reliably (https://doi.org/10.1016/j.ejim.2021.11.018).   We have now specified this point in the Materials and Methods section, second paragraph, according to the reviewer’s suggestion.

Reviewer 2 Report

Comments and Suggestions for Authors

Metformin has shown reduced mortality and overall beneficial effects in many prospective and retrospective clinical studies earlier. However, controversial reports have been shown in several studies in COVID-19-infected diabetic patients. In agreement with other studies, this report also showed overall lower mortality in the metformin-treated group.

1. The study design and analysis are robust and have considered several co-morbidities. However, whether the metformin group was on antiviral medication during their hospital stay is not discussed or not considered in multivariate analysis.  Antiviral treatment may be effective in lowering the mortality in bivariate analysis.

2. Table 1, no. of co-morbidity is similar in all groups, the P value should not be 0.05. Please recheck.

3. Line 233, check the typo covid.

4. Line 224   id test, check, and explain

5. Line 174 Low o, correct it.

6. Statistical analysis needs to separate sections under materials & methods sections for better clarity 

7. Table needs a legend to show the comparison between the two groups.

Author Response

Metformin has shown reduced mortality and overall beneficial effects in many prospective and retrospective clinical studies earlier. However, controversial reports have been shown in several studies in COVID-19-infected diabetic patients. In agreement with other studies, this report also showed overall lower mortality in the metformin-treated group.

  1. The study design and analysis are robust and have considered several co-morbidities.

We thank the reviewer for her/his consideration

However, whether the metformin group was on antiviral medication during their hospital stay is not discussed or not considered in multivariate analysis.  Antiviral treatment may be effective in lowering the mortality in bivariate analysis.

We thank the reviewer for her/his criticism. At the time in which most of the patients were enrolled in the study, few efficacious antiviral medications were available: in particular, in our study, some patients were prescribed Ritonavir + Lopinavir or darunavir+cobicistat, which have been proven to have limited efficacy for COVID-19 treatment: we have now performed a bivariate analysis, showing a no significant correlation between antiviral medications and mortality. [OR 2.44       (0.97-6.14), P= 0.06]: furthermore, when adding antviral use to the multivariate model together with metformin, previous heart disease and COVID-MRS, neither antiviral use              [OR1.89              (0.64-5.62) P= 0.25] or metformin use [OR 0.71              (0.27-1.9), P= 0.50] showed a significant correlation with mortality. We have now added these analyses to the results section, and we briefly discussed them in the discussion section.

  1. Table 1, no. of co-morbidity is similar in all groups, the P value should not be 0.05. Please recheck.

We apologize for previous mistake. P-Value was 0.11. We have now corrected the table.

  1. Line 233, check the typo covid.

We apologize for our previous inaccuracy: The typo has now been corrected.

  1. Line 224   id Test, check, and explain

“id est” is a latin conjunction, sometimes used in English as a synonym for “in other words”. We have now rephrased the sentence for better clarity.

  1. Line 174 Low o, correct it.

We apologize for our previous mistake, which has now been corrected according to your suggestion

  1. Statistical analysis needs to separate sections under materials & methods sections for better clarity

We thank the reviewer for her/his careful remarks: we have now created separate sections under material and methods, for population, endpoints, data collection and Statistical analysis.

  1. Table needs a legend to show the comparison between the two groups.

We thank the reviewer for her/his criticism: the legend was present under the table, however we added some information to the table, for greater clarity.
